# Effects of Different Sources of Culture Substrate on the Growth and Immune Performance of the Red Swamp Crayfish (*Procambarus clarkii*)

**DOI:** 10.3390/ijms241814098

**Published:** 2023-09-14

**Authors:** Rong Wan, Chengfeng Zhang, Yongkai Tang, Jian Zhu, Nan Yang, Shengyan Su

**Affiliations:** 1Wuxi Fisheries College, Nanjing Agricultural University, Wuxi 214128, China; wrdaytoy@163.com (R.W.); zhangcf@ffrc.cn (C.Z.); tangyk@ffrc.cn (Y.T.); zhuj@ffrc.cn (J.Z.); 15696141341@163.com (N.Y.); 2Lab of Natural Food and Fish Culture, Freshwater Fisheries Research Center, Chinese Academy of Fishery Sciences, Wuxi 214128, China

**Keywords:** *Pracambarus clarkii*, cultured substrate, growth performance, antioxidant capacity, intestinal microbiota

## Abstract

The substrate in the aquatic environment plays a crucial role in nutrient deposition and recovery for the growth of aquatic organisms. In order to optimize the culture medium of Procambarus Clarkii, culture media from different sources were selected in this study to explore their effects on the growth and immune performance of red swamp crayfish. The results showed that the weight gain rate (WGR), body length growth rate (BLGR) and specific growth rate (SGR) in group I2 were the highest, followed by group I1 and group I3. The WGR and SGR of crayfish in the I1 and I2 groups were significantly higher than those in the I3 group (*p* < 0.05). The activities of acid phosphatase (ACP), alkaline phosphatase (AKP) and superoxide dismutase (SOD) were the highest in group I2, followed by group I3, and the lowest in group I1. The expression trends in growth-related genes, nuclear hormone receptor (E75), molt-inhibiting hormone (MIH) and chitinase genes were similar, and the expression levels in the I2 group were higher than those in the I1 and I3 groups. It was noted that the expression levels of E75 and MIH genes in the I2 group were significantly higher than those in the I3 group (*p* < 0.05). α diversity analysis of 16S rRNA data showed that there was no statistically significant difference in the abundance of intestinal flora among the three culture substrate groups. The β diversity in the Xitangni group, crayfish Tangni group and Shuitangni group was significantly different. These changes in microbiota suggest that using different substrates to culture crayfish leads to differences in gut microbiota diversity. To sum up, the growth in crayfish and immune performance influenced by the culture substrate condition and aquatic breeding sediment substrates, rather than crab pool and paddy field pond sediment substrates, showed a better effect.

## 1. Introduction

*Procambarus clarkii*, commonly known as crayfish or red crayfish, is one of the most important farmed freshwater crayfish species worldwide [1,2]. It is native to southeastern North America and was introduced to China in the 1930s. Over the years, China’s crayfish aquaculture area and output have shown a consistent upward trend. According to the Crayfish Industry Report 2023 (http://www.moa.gov.cn/, accessed on 12 June 2023), in 2022, the farming area of crayfish in China reached 28 million mu (1.87 million hectares), with a production of 2.89 million tons, accounting for 8.79% of the total freshwater aquaculture production in the country. This placed crayfish as the fourth-most-produced freshwater aquaculture species in China for the first time. There were a total of 24 provinces in China that reported crayfish farming production. Among them, the five traditional major crayfish farming provinces, which are Hubei, Anhui, Hunan, Jiangsu and Jiangxi, still maintained the dominant position, with a farming production of 2.6374 million tons, accounting for 91.24% of the total crayfish farming production in the country. Crayfish farming primarily includes various methods such as rice–crayfish co-cultivation, pond farming, large water surface extension farming and farming in lotus and water chestnut fields. Among them, rice–crayfish co-cultivation covers an area of 23.5 million mu (1.57 million hectares), accounting for 83.93% of the total crayfish farming area. Pond farming covers an area of 3.5 million mu (233,333 hectares), accounting for 12.50%. Pond farming includes mixed farming of crayfish and crabs, as well as specialized crayfish pond farming, with mixed farming being the main practice.

Aquaculture pond sediments hold significant ecological importance as habitats that are often subject to disturbance. Zhang et al. [3] conducted a study aimed at examining the influence of fish and crab farming practices on bacterial communities and enzymatic activities within sedimentary environments. The results indicated that crab ponds exhibited significantly greater alpha diversity in sediment when compared to fish and crayfish ponds. Additionally, the study revealed a strong correlation between the diversity and structure of microbial communities and sediment properties, while the distribution and composition of fungal communities were primarily influenced by species-specific farming practices. Zhang et al. [4] discovered that the farming practices employed for fish and crab cultivation have distinct impacts on both bacterial communities and enzymatic activities within pond sediments. These findings provide evidence for the influence of nutrient-driven sediment biological processes in aquaculture ponds, which vary depending on the specific farming practices implemented.

In crustaceans, humoral immunity is primarily regulated by enzymes, including alkaline phosphatase (AKP) [5], hemolymphatic acid phosphatase (ACP) [6], catalase (CAT) [7] and superoxide dismutase (SOD) [8]. These enzymes can be used as indexes to evaluate nonspecific immunity as well [9,10]. The enzymes AKP and ACP regulate metabolism in crustaceans and play an essential role in improving disease resistance and substance metabolism [9,11,12].

The *E75* gene is an ecdysteroid-inducing gene that plays a crucial role in the molting process of arthropods. The protein that it encodes serves as the primary target for the ecdysteroid receptor (EcR) [13]. The *MIH* gene is an ecdysteroid inhibitory gene. It exerts an inhibitory effect on molting by suppressing the synthesis and secretion of ecdysteroids by its target organ, the Y organ, not only during the pre-ecdysis period [14]. Chitinase, a protein widely present in animals, plants, bacteria and fungi, serves multiple functions in crustaceans. It aids in the breakdown of chitin found in their exoskeletons and food, while also playing a role in the defense against pathogen invasion [15].

In this experiment, three different crayfish culture methods—crab and shrimp poly-culture, pond culture and paddy culture—were examined. Aquaculture pond sediments were chosen as the variable, with crab pond mud, crayfish pond mud and paddy pond mud used as substrates for different experimental groups, while other test conditions remained consistent. The study investigated the effects of different substrate sources on the growth performance and health status of crayfish by comparing four growth indexes (body length growth rate, weight gain rate, specific growth rate and survival rate), four oxidation indexes (hemolymphatic acid phosphatase, alkaline phosphatase, catalase and superoxide dismutase), the composition and differences in intestinal flora in each group and the expression of genes related to hepatopancreatic growth.

## 2. Results

### 2.1. Growth Performance

The weight gain rate (WGR) and specific growth rate (SGR) differed among the three experimental groups, with the crayfish pond mud group (I2 group) exhibiting the highest rates, followed by the crab pond mud group (I1 group) and then the paddy pond mud group (I3 group). Both the weight gain rate (WGR) and specific growth rate (SGR) of the I1 group and I2 group were significantly higher than those of the I3 group (*p* < 0.05). However, there was no significant difference in body length growth rate (BLGR) and survival rate (SR) observed among the three groups (Table 1).

### 2.2. Serum Antioxidant Indicators 

After conducting the culture test, the serum samples were extracted from the crayfish to determine the immune indexes. The results indicated that the activities of ACP (Figure 1a), AKP (Figure 1b) and SOD (Figure 1d) were the highest in the crayfish pond mud group, followed by the paddy pond mud group and the crab mud group. Notably, the AKP activity in the serum of crayfish from the crayfish pond mud group was significantly higher than that in the crab mud group (*p* < 0.05).

Furthermore, the CAT (Figure 1c) activity followed a descending order, with the crab pond mud group exhibiting the highest activity, followed by the paddy pond mud group and the crayfish pond mud group. The CAT activity in the crab pond mud group was significantly higher than that in the crayfish pond mud group (*p* < 0.05). No significant differences were observed in the enzyme activities of AKP and CAT in the serum of crayfish between the paddy pond mud group and the other two groups. Similarly, no significant differences were found in the ACP and SOD activities in the serum of crayfish (Figure 1).

### 2.3. Intestinal Flora

The alpha diversity analysis of the intestinal flora in each group of crayfish was conducted. In the analysis of the rarefaction curve, it was observed that as the number of reads increased, the alpha diversity index of each sample reached a plateau (Figure 2a). This suggests that the sequencing data quantity was appropriate, adequately covering the entire intestinal flora of crayfish. Consequently, it accurately and effectively reflected the diversity in the intestinal flora, making it suitable for subsequent analysis. In the analysis of the rank abundance curve, the intestinal microbiota in the crayfish from the three groups was investigated and these curves provide insights into both species abundance and distribution evenness (Figure 2b). The presence of a broader range on the horizontal axis is indicative of a greater abundance of species. The smoothness of the curve serves as a reflection of the uniformity in the distribution of species within the sample, with a gentler curve suggesting a more even distribution. It is evident that the curves in all three groups exhibited similar inclinations and widths, suggesting no significant disparities in species richness and uniformity among them. Furthermore, no significant differences were observed in the Chao1, Shannon and Simpson indices among the three groups, indicating a lack of variation in alpha diversity among the aforementioned groups (Table 2).

The samples on the coordinate axis were relatively close, indicating similar species composition within each group that could not be distinguished (Figure 3). However, beta diversity differences between groups could still be analyzed using the Student t-test. According to the Weighted Unifrac Distance, there was a highly significant difference between the crab mud group and the paddy pond mud group (*p* < 0.01), as well as between the crayfish pond mud group and the paddy pond mud group (*p* < 0.001). Similarly, based on the Unweighted Unifrac Distance, there was a highly significant difference between the crab pond mud group and the crayfish pond mud group (*p* < 0.001), as well as between the crayfish pond mud group and the paddy pond mud group (*p* < 0.0001) (Figure 4). In summary, there were significant differences in beta diversity among the crab pond mud group, crayfish pond mud group and paddy pond mud group. These differences in microbiota indicate that the cultivation of crayfish using different substrates leads to variations in the diversity of intestinal flora.

The top 10 microorganisms at the phylum level in the intestinal abundance of crayfish cultured in the three experimental groups were identified as Firmicutes, Bacteroidota, Proteobacteria, Chloroflexi, Patescibacteria, Actinobacteriota, Acidobacteriota, Cyano-bacteria, Planctomycetota and Verrucomicrobiota. Among these, Firmicutes, Bacteroidota and Proteobacteria were found to be the dominant flora in all three groups, constituting 99.24%, 97.74% and 99.53% of the intestinal flora of crayfish cultured in the three experimental groups, respectively (Figure 5a). There was no statistically significant disparity observed in the composition and abundance of intestinal microflora across the three experimental groups at the phylum level. Nevertheless, the abundance of Firmicutes was positively correlated with weight gain rate (WGR), body length growth rate (BLGR) and specific growth rate (SGR), while the abundance of Bacteroidota displayed a negative correlation.

The top 10 abundant genera are Bacteroides, RsaHf231, Anaerorhabdus furcosa group, Shewanella, Dysgonomonas, Aeromonas, Candidatus Bacilloplasma, ZOR0006, Tyzzerella and Niveibacterium. Among them, the abundance of Bacteroides, RsaHf231, Anaerorhabdus furcosa group, Shewanella and Dysgonomonas were found to be the dominant flora in all three groups, constituting 50.89%, 55.42% and 60.3% of the intestinal flora of crayfish cultured in the three experimental groups, respectively (Figure 5b). Meanwhile, the abundance of Bacteroides was negatively correlated with weight gain rate (WGR), body length growth rate (BLGR) and specific growth rate (SGR), while the abundance of Bacteroides displayed a positive correlation. Among the three experimental groups, there were no statistically significant differences in the composition and abundance of intestinal microflora at the genus level.

An evolutionary branch diagram of microbes in the intestine was produced using the LEfSe algorithm analysis (Figure 6a). The results showed that the crab pond mud group (I1 group) was the most differentially abundant group.

Through the analysis of microorganisms with an LDA value < 4 in each group (Figure 6b), significant intergroup differences were observed in the species taxonomies from the order level to the genus level, encompassing a total of 11 species taxonomies, including 2 orders, 4 families and 5 genera. At the order level, the relative abundance of Acetobacterales in the crayfish pond mud group (I2 group) was significantly higher compared to the other groups, while the relative abundance of Lachnospirales in the paddy pond mud group (I3 group) was significantly higher as well. Moving to the family level, the relative abundance of Aeromonadaceae and env_OPS_17 in the crab pond mud group (I1 group) was significantly higher than in the other groups. At the genus level, the relative abundance of SH_PL14, Edaphobaculum and Chryseobacterium in the crab pond mud group (I1 group) was significantly higher than in the other groups (Figure 6).

Based on the KEGG database, the potential functions of the intestinal flora of crayfish cultured in different pond sediment groups were predicted using PICRUSt2 2.4.2 software. The functional genes predicted by the intestinal flora of crayfish can be classified into six Level 1 functional pathways and forty-two Level 2 functional pathways in the KEGG database. These pathways include 11 pathways in Metabolism, 4 pathways in Genetic Information Processing, 3 pathways in Environmental Information Processing, 5 pathways in Cellular Processes, 9 pathways in Organismal Systems and 10 pathways in Human Diseases. Notably, there were no significant differences in gene function prediction among the intestinal samples from the three experimental groups (*p* > 0.05). This suggests that the intestinal gene function of crayfish remains consistent regardless of the substrate conditions under which they are grown (Figure 7).

### 2.4. Growth-Related Gene Expression

The expression of the *EcR* gene in the hepatopancreas of *Procambarus larkia*, with AK as the internal reference gene, followed the following pattern: paddy pond mud group > crayfish pond mud group > crab pond mud group. Furthermore, the expression of the *EcR* gene in the crab pond mud group was significantly higher than that in the paddy pond mud group (*p* < 0.01). Regarding the *E75*, *MIH* and *chitinase* genes, their expressions were observed as follows: crayfish pond mud group > crab pond mud group > paddy pond mud group. Notably, the expression of the *E75* and *MIH* genes in the crayfish pond mud group was significantly higher than that in the paddy pond mud group (*p* < 0.05). However, no significant difference was found in the expression of the chitinase gene among the three groups (*p* > 0.05) (Figure 8).

### 2.5. Correlation Analysis of Growth and Immune Indexes, Growth Indicators and Growth-Related Gene Expression

In terms of growth indexes, the weight gain rate showed a significant positive correlation with the growth rate of body length (*p* < 0.05) and a very strong positive correlation with the specific growth rate (*p* < 0.01), and the growth rate of body length was significantly positively correlated with the specific growth rate (*p* < 0.05). Regarding immunological indexes, AKP activity exhibited a significant negative correlation with CAT activity (*p* < 0.05) (Table 3). For growth gene expression, *MIH* and *E75* expression demonstrated a significant positive correlation (*p* < 0.05) (Table 4). No significant correlations were found for the remaining indicators.

Regarding both growth indexes and immune indexes, ACP and AKP activity showed positive correlations with weight gain rate, body length growth rate and specific growth rate, while exhibiting negative correlations with survival rate. CAT and SOD activity displayed negative correlations with weight gain rate, body length growth rate and specific growth rate, but showed positive correlations with survival rate. In terms of growth indexes and growth gene expression, *EcR* gene expression was negatively correlated with weight gain rate, body length growth rate and specific growth rate, but positively correlated with survival rate. On the other hand, *E75*, *MIH* and *chitinase* gene expression exhibited positive correlations with weight gain rate, body length growth rate and specific growth rate, while they were negatively correlated with survival rate. No significant correlations were observed between growth indexes and immune indexes, as well as between growth indexes and growth-related gene expression (*p* > 0.05).

## 3. Discussion

In China, crayfish have gained prominence as the most productive freshwater shrimp due to their substantial commercial value. Traditional pond cultivation and emerging rice co-culture methods are the primary approaches employed for crayfish breeding, with integrated rice–crayfish co-culture being recognized as a successful example of integrating livestock and crop production within eco-agricultural systems. 

It has been shown in previous studies that pond culture greatly influences the environmental conditions of pond sediments in the agricultural sector [16]. The utilization and widespread adoption of commercial feed have led to the introduction of a significant quantity of nutrient substances into ponds, ultimately resulting in their deposition on the pond’s bottom [3]. Bacteria play a crucial role in the decomposition and reutilization of fundamental elements, including carbon, nitrogen and phosphorus, within sediments [17]. It was found that with the increase in breeding years, the contents of total nitrogen and organic matter increased significantly, and the oxidation–reduction potential and mean sediment grain size decreased significantly [18]. The introduction of nutrients through agricultural practices has the potential to modify sediment properties, thereby influencing the composition of bacterial communities. These communities are crucial in regulating the aquatic environment and sediment function. Furthermore, over extended periods of aquaculture, for example, more than 15 years, sediments derived from ponds housing the same cultured species may exhibit greater similarity in their bacterial communities [4]. Sediment properties change with culture years and ultimately influence the output to input ratio of ponds and microbes [18]. 

Multiple studies have shown that the properties of pond sediments are influenced by farming practices, including resulting in differences in microbial communities and differences in enzyme activity, and that this effect is relatively stable and long-lasting [3,4]. A study found that the developmental stage and then feeding were better determinants of gut microbiome patterns of crayfish than geography [19]. A multi-domain analysis of a variety of co-culture models for rice, crayfish and mitten crabs showed that the bacterial, archaea, viral and eukaryotic communities in bottom sediments showed better stability than those in the gastrointestinal tract of crayfish and mitten crabs [20]. 

This study investigated three widespread crayfish culture techniques, namely crab–crayfish co-culture, single crayfish culture and rice–crayfish co-culture, and chose the bottom sediments of aquaculture ponds that had been farmed for five years as substrates for the experimental groups. All other test conditions were kept constant. As a result of the study, crayfish cultured in crayfish pond mud demonstrated the best growth and immunity performance, followed by those cultured in crab pond mud and paddy pond mud. Furthermore, we conducted an investigation into the composition of the intestinal microbiota in crayfish, considering three distinct substrate conditions. Notably, we observed notable disparities in microbial diversity and the relative abundance of species among the three groups. Specifically, there was a highly significant discrepancy in beta diversity between the aforementioned groups (*p* < 0.01). 

The presence and proportion of Firmicutes and Bacteroidota microbiota, two prominent microbial communities in the gastrointestinal tract, have substantial implications for the well-being and metabolic functions of the host [21,22]. Specifically, in healthy adult individuals, Firmicutes and Bacteroidetes account for over 90% of the microbiota, maintaining a relatively consistent state and participating in the metabolism of plant-derived polysaccharides [23,24]. A higher abundance of Firmicutes compared to Bacteroidota enables the host to efficiently absorb and utilize the energy derived from food [25]. Members of Firmicutes are beneficial microorganisms in agroecology and are often applied to promote plant growth and bioremediation [16]. The Proteobacteria exhibit dominance within the gut microbiotas of Crustacea, particularly in aquatic invertebrates, and demonstrate considerable diversity in terms of physiology, morphology and genetics [26].

In the present study, it was observed that Firmicutes exhibited the highest abundance among the three groups, aligning with previous findings in red crayfish, where Firmicutes were notably enriched in sediments of newly formed ponds (with a culture period of less than one year). Additionally, the Bacteroidota and Proteobacteria rank as the second- and third-most-prevalent intestinal microflora in all three groups. In line with prior investigations, the consumption of fermented feed is shown to substantially elevate the abundance of Bacteroidetes, while concurrently reducing the abundance of Proteobacteria in adult crayfish [19]. 

The Bacteroides group holds significant importance within the intestinal microbiota of Crustacea [27]. This group possesses numerous genes associated with polysaccharide and monosaccharide metabolism, thereby playing a crucial role in nutrient absorption in crayfish [28,29]. Furthermore, Bacteroides is capable of producing propionate, which has the potential to alleviate colitis and enhance the intestinal barrier function while reducing inflammation [30]. The inclusion of fermented feed comprising soybean, corn and wheat in the diet provides a greater supply of polysaccharides and probiotics (yeast and lactic acid bacteria) compared to the basal diet. Crayfish fed fermentation-supplemented diets exhibited an increase in Bacteroidetes in their gut microbiota. Thus, the gut microbiota of crayfish supplemented with fermented feed has an increased amount of Bacteroidetes. Furthermore, RsaHF231 has been identified in the gut microbial community of crayfish, indicating its transient nature and a decline in relative abundance as development progresses. However, the biological significance of the phylum RsaHF231 remains unclear, necessitating further investigation to provide a comprehensive and meaningful explanation. In our study, it was observed that the crayfish belonging to the crab mud group with the highest growth rate exhibited the lowest abundance of Bacteroides. Conversely, the crayfish in the rice mud group with the poorest growth rate displayed the highest Bacteroides abundance. This finding implies that the crayfish in the crab mud group consumed organic nutrients present in the sediment, in addition to the fermented feed, whereas the crayfish in the rice mud group consumed a greater proportion of fermented feed. At the genus level, apart from Bacteroides, RsaHF231 was identified as the second-most-abundant microbiome in all three groups of crayfish. However, no definitive conclusions can be made based on the analysis. Previous studies have suggested that the Anaerorhabdus furcosa group, the third-most-abundant microbiome in all three groups of crayfish, may be associated with oxidative stress in crayfish, a hypothesis supported by serum antioxidant indexes [31,32,33]. These findings suggest that the intestinal flora of the crab pond mud group is more imbalanced, with a weaker immune system and potentially higher susceptibility to diseases. This observation is consistent with the decreased levels of crayfish serum ACP, AKP and SOD activity and lower survival rate (although not significantly different) observed in the group exposed to crab pond mud.

The analysis of the intestinal flora of crayfish cultured in various pond sediment groups revealed no statistically significant variations in gene function prediction among the intestinal samples from the three experimental groups (*p* > 0.05). This implies that the intestinal gene function of crayfish remains stable irrespective of the substrate conditions in which they are cultivated.

The rice mud group demonstrated the highest expression of the *EcR* gene, whereas the *MIH* expression was observed to be the lowest. In contrast, the crayfish pond mud group displayed elevated expression levels of molt genes and shell inhibitory genes, indicating a heightened molting activity in the shrimp belonging to this particular group.

In summary, our study demonstrates significant differences in the intestinal microbial composition of crayfish under different substrate conditions. These findings suggest that the microbial communities present in sediment within crayfish ponds exhibit significant variations depending on the employed culture techniques, potentially influencing the overall efficiency of cultivation. The results of this study indicate that the incorporation of advantageous microorganisms into production practices can be accomplished by introducing pond bottom mud from crayfish that have been separately cultured in the traditional rice–shrimp farming method. This approach is anticipated to enhance crayfish yield. Furthermore, the study suggests that during critical stages of crayfish cultivation, such as egg, larval and pre-adult phases, it may be feasible to initially rear crayfish in a pond bottom mud environment with a relatively diverse microbiome, followed by the adoption of a mixed rice–shrimp culture model. 

## 4. Materials and Methods

### 4.1. Culture Experiment

The experiment was conducted between August and September 2022 at the Bait Biological Culture and Breeding Laboratory of Wuxi Fishery College, which is affiliated with Nanjing Agricultural University (located at 120.250479° E, 31.51581° N; Wuxi, China). The experimental setup consisted of nine glass tanks, equipped with an aeration system, as well as an inlet and drainage system. Experimental glass tanks were filled with filtered freshwater after being coated with a 5 cm layer of sediment. The experimental sediment was the bottom mud of the crab pond, crayfish pond and paddy pond, respectively. Each breeding pond had had three years of breeding experience before, with normal feeding, no disease and water quality testing often being carried out, meeting the standard requirements. The chemical composition of the three substrates was at the normal level, and the composition of the microbial community was mainly formed of *Firmicutes*, *Proteobacteria*, *Bacteroidetes*, *Chloroflexi*, *Acidobacteria*, *Actinobacteria*, *chloroflexi*, *Patescibacteria*, *Verrucomicrobia* and *BRC1*. All of the sediment was collected from ponds that had been farmed for five years. An equal amount of sediment samples were collected from each pond at three random locations. The sediment samples from the same pond were diluted in 500 mL of sterile water and filtered through a 100 µm mesh sieve. After collection, all sediment samples were instantly sent to the laboratory and settled in the corresponding tanks for three days before experimentation. 

Throughout the duration of the experiment, the freshwater used was tap water that had undergone aeration, resulting in a pH of 7.12 ± 0.07 and an ammonia–nitrogen concentration of 0.10 ± 0.02 mg/L. The water temperature was maintained at 27.5 ± 1.5 °C, while the dissolved oxygen level was kept at 6.0 ± 0.2 mg/L. To ensure water quality, one third of the water was replaced every two days. The lighting conditions in the experiment followed a natural light and dark cycle.

The shrimp used in the experiment were sourced from a specialized aquatic seed farm located in Yueqi, Yangshan Town, Wuxi City. To adapt to laboratory conditions, these shrimp were put in a glass tank for 14 days before the experiment. Ninety young shrimp had similar body weights (9.55 ± 0.81 g), and all of the shrimp exhibited no signs of mortality and maintained good health throughout the duration of the experiment. The experimental diet consisted of commercial feed (Changzhou Wanfeng Feed Co., Ltd., Changzhou, China) that was finely ground and filtered through a 100 µm mesh sieve. This feed contained a minimum crude protein content of 42%, a maximum crude fiber content of 6%, a minimum crude fat content of 5%, a minimum lysine content of 2%, a minimum total phosphorus content of 1.2% and a maximum water content of 13%. Feeding occurred twice a day at 9:00 and 16:00 h, adhering to a predetermined feeding regimen.

### 4.2. Sample Collection

The body length and weight of each shrimp group were measured before and after the culture test. Following the culture test, the shrimp were fasted for one day. Subsequently, samples of the hepatopancreas, serum and intestinal contents were collected from each shrimp group and stored in a −80 °C freezer. These samples were later used to determine the expression of growth-related genes, antioxidant indexes and intestinal flora, respectively.

### 4.3. Growth Index Determination

During the experiment, several parameters were measured for crayfish, including the initial and final length, initial and final weight, initial number, survival quantity and feed intake. To evaluate the growth performance, four indicators were chosen: body length growth rate, weight gain rate, specific growth rate and survival rate. The calculation formulae for these indicators are as follows:Body length growth rate (BLGR, %) = (L_t_ − L_0_)/L_0_ × 100;
Weight gain rate (WGR, %) = (W_t_ − W_0_)/W_0_ × 100;
Specific growth rate (SGR, %/d) = (lnW_t_ − lnW_0_)/t × 100;
Survival rate (SR, %) = X_t_/X_0_ × 100;
whereby L_0_ is the average initial body length (cm) of crayfish, L_t_ is the average final body length (cm) of crayfish, W_0_ is the average initial body weight of crayfish (g), W_t_ is the average final body weight of crayfish (g), t is the number of days of rearing, X_t_ is the survival quantity of crayfish at the end of the culture trial and X_0_ is the initial number of crayfish at the beginning of the culture trial.

### 4.4. Determination of Antioxidant Indexes

Acid phosphatase (ACP) [34], alkaline phosphatase (AKP) [5], catalase (CAT) [35] and superoxide dismutase (SOD) [7,9] were selected as evaluation indexes for assessing the antioxidant capacity of a crayfish’s blood lymph [6,8,10,36,37]. To begin, the surface water on the shrimp’s body was carefully wiped off using a paper towel. Subsequently, a sterile 1 mL syringe was inserted at the midpoint between the shrimp’s cephalothoracic and ventral boundaries to collect blood. The collected blood lymphatic fluid was then transferred to a sterile centrifuge tube. Anticoagulant, in a 4× volume relative to the lymphatic fluid, was added to the centrifuge tube. Following an overnight incubation at 4 °C, the tube was centrifuged at 5000 rpm/min for 15 min, and the supernatant was collected for further measurement. The measurement kit used in this study was obtained from the Nanjing Jiancheng Institute of Bioengineering, and the measurement procedures were conducted according to the provided instructions.

### 4.5. Intestinal Flora Determination 

After conducting the culture test, three shrimp were randomly selected from each tank. The selected shrimp were then dissected in an ice bath within a sterile environment, and their intestines were carefully extracted. The extracted intestines were placed in 2 mL cryopreservation tubes and stored at −80 °C for the subsequent analysis of intestinal microbes. In total, there were 6 intestinal samples in the crab mud group and 9 intestinal samples in each group of the crayfish pond mud and paddy pond mud, resulting in a total of 24 intestinal samples. The sequencing of the samples was performed at Nanjing Jisihuiyuan Biotechnology Co., Ltd., Nanjing, China. 

### 4.6. Bioinformatic Analysis

To enhance the precision and dependability of outcomes in subsequent bioinformatic analysis, the raw data underwent preprocessing, utilizing an internally developed procedure. The software Pandaseq 2.11 [38] was employed to merge paired-end reads into a unified sequence by leveraging their overlapping relationship. Furthermore, the software PRINSEQ 0.20.4 [39] was employed to filter out bases with quality values below 20 at the tail end of reads and discard sequences where the length of “N” represented 5% of the total sequence length.

The calculation of alpha diversity [40] involved two steps. Firstly, dilution curve analysis was conducted using R (V3.6.2) to plot dilution curves for various alpha diversity indices, with the alpha diversity index value corresponding to the randomly extracted sequencing data size. Secondly, sample complexity analysis was performed using R (V3.6.2) to conduct rank sum test scores for different alpha diversity indices. The diversity difference between groups was then analyzed and represented using a box diagram.

The beta diversity analysis involved two components. Firstly, beta diversity index statistics were calculated using the software QIIME2 2021.11 (https://github.com/QIIME2/q2-feature-classifier, accessed on 2 September 2023), utilizing the relative abundance of amplicon sequence variant (ASV) and employing Weighted Unifrac and Unweighted Unifrac Distances. The resulting diversity index box plot was generated using the R (V3.6.2) vegan package anosim function. Secondly, the PCoA analysis diagram was constructed using the Python (V2.7.18) matplotlib library, based on the two aforementioned distance matrices.

The analysis of species community structure involved examining the abundance and annotation information of amplicon sequence variant (ASV). Statistical analysis was conducted to determine the proportion of sequences in each sample at various classification levels, thereby evaluating the resolution of species annotation and the complexity of the sample’s species composition. Version 3.6.2 was employed to generate a stacked bar chart illustrating the abundance of the top 20 species across different taxonomic levels.

LefSe was used to analyze species differences between LEfSe groups in this study by utilizing the relative abundance of amplicon sequence variant (ASV) as a biometric identifier. Additionally, an evolutionary branching diagram and a histogram of LDA value distribution were constructed (https://github.com/biobakery/lefse/archive/1.0.0.tar.gz, accessed on 2 September 2023).

Function prediction was performed using PICRUSt2 software (V2.1.2) [41] by matching ASV representative sequences to the KEGG PATHWAY database (https://www.kegg.jp/kegg/pathway.html, accessed on 2 September 2023) and COG database (https://www.ncbi.nlm.nih.gov/research/cog-project/, accessed on 2 September 2023). This allowed for the forecast of ASV function and the determination of function abundance spectrum. The functional abundance stacked bar charts and COG annotated classification bar charts at different classification levels of KEGG were drawn using the R (V3.6.2) ggplot2 program package and Perl (V5.28.3) SVG module, respectively.

### 4.7. Growth-Related Gene Determination

After the culture test, five crayfish individuals were randomly selected from each group. They were dissected in a sterile environment within an ice bath, and the hepatopancreas was placed in a 2 mL cryopreservation tube. The samples were then stored at -80 °C for subsequent index determination. The total RNA was extracted from the liver and pancreatic tissues of crayfish using Trizol (RNAiso Plus, TaKaRa, Beijing, China), and its OD value and concentration were measured using a spectrophotometer.

For reverse transcription, the HiScript III RT SuperMixfor qPCR (+gDNA wiper) kit was used according to the provided instructions. The resulting product was stored in a -20 °C freezer for subsequent experiments. Aspartate kinase (*AK*) was chosen as the internal reference gene [9] to validate the expression of four growth genes, namely ecdysone receptor (*EcR*), nuclear hormone receptor (*E75*), molt-inhibiting hormone (*MIH*) and *chitinase*, in the liver and pancreatic tissues of crayfish. The primer sequences were synthesized by Shanghai Tianlin Biotechnology Co., Ltd., Shanghai, China, and their details can be found in Table 5. The qPCR reaction was performed using the ChamQ Universal SYBR qPCR Master Mix kit. The relative expression of *EcR*, *E75*, *MIH* and *chitinase* genes was calculated using the 2^(−ΔΔCT)^ method.

### 4.8. Statistical Analysis

The data are presented as the mean ± standard deviation, unless otherwise specified in the figure legends. The figure legends also indicate the sample number (n), which represents the number of crayfish samples used in each experiment. Prior to analysis, the data of the growth index, antioxidant index and expression of growth-related genes were assessed for normality using the Shapiro–Wilk test of normality and for homogeneity of variance using Levene’s test for homogeneity of variance. In cases where the data in crayfish satisfied the assumptions of the parametric statistical tests, the results were subjected to analysis using Student’s t test. Meanwhile, correlation analysis between growth indices and growth-related gene expression was adopted using Pearson’s correlation.

The statistical details of the experiments can be located in Section 2, where differences in means were deemed to be statistically significant at a significance level of *p* < 0.05. The analyses were conducted using IBM SPSS Statistics software (version 24), while GraphPad Prism 9 was employed for the creation of the graphs.

## 5. Conclusions

An analysis was conducted on the growth indexes, serum immunity indexes, intestinal flora and growth gene expression of crayfish cultured in different sources of breeding substrate. The results revealed that crayfish cultured in crayfish pond mud demonstrated the best growth and overall immune performance. Additionally, the relative abundance of acetic acid bacteria in the intestine was higher compared to other groups. Crayfish cultured in crab pond mud showed the second-best growth, but three out of the four measured immunoenzyme activities were lower than those of the other groups. Pathogenic bacteria, such as *Aeromonas* and *Flabobacterium aureus*, were also found in the intestine of this group. These findings suggest that shrimp in this group had the weakest immunity and were more susceptible to diseases. Crayfish cultured in the paddy pond mud group exhibited the poorest growth, followed by the next best overall immune performance. No significant correlation was observed between the growth indexes and immune indexes, as well as between growth indexes and the growth gene expression of crayfish in each group. In summary, the optimal condition for cultivating crayfish is with pond mud as the substrate.

## Figures and Tables

**Figure 1 ijms-24-14098-f001:**
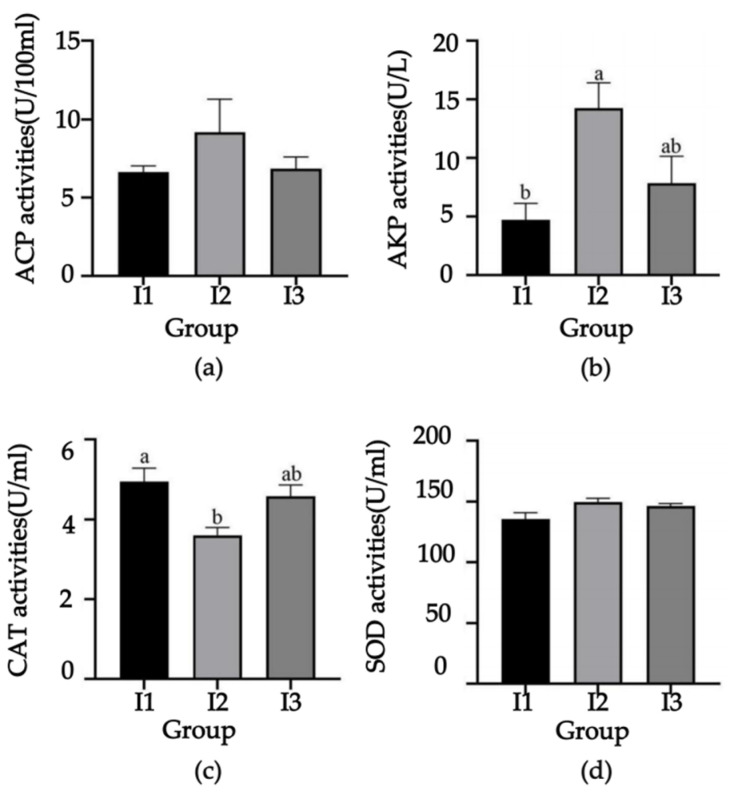
Effects of different sources of culture substrate on serum antioxidant indexes of crayfish. (**a**) Acid phosphatase (ACP) activity; (**b**) alkaline phosphatase (AKP) activity; (**c**) catalase (CAT) activity; (**d**) superoxide dismutase (SOD) activity. I1, crab pond mud group; I2, crayfish pond mud group; I3, paddy pond mud group. Different letters in the same column in the table indicate significant differences (*p *< 0.05), and no significant differences with the same letters or no letters (*p *> 0.05). The same as below.

**Figure 2 ijms-24-14098-f002:**
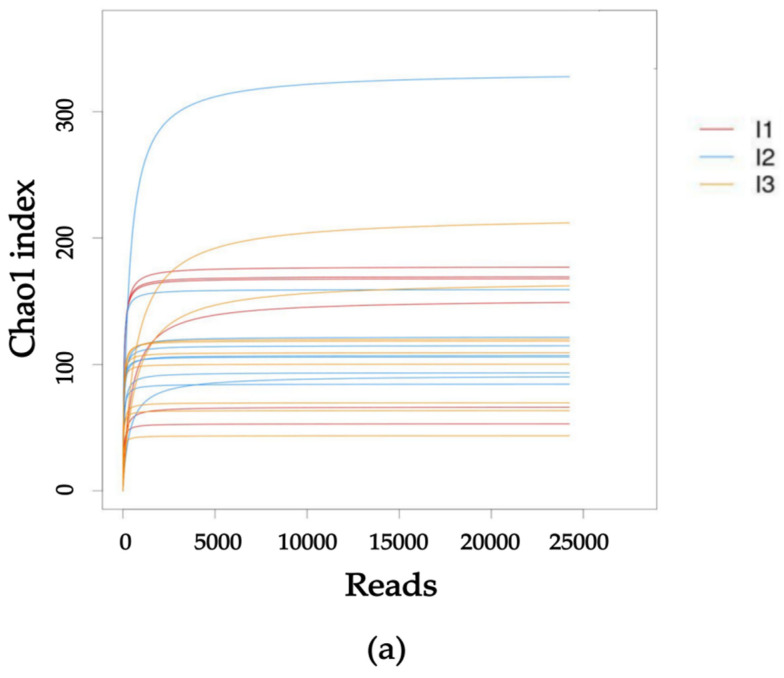
Analysis of alpha diversity dilution curves and grade abundance curves. (**a**) Rarefaction curve; (**b**) rank abundance curve. I1, crab pond mud group; I2, crayfish pond mud group; I3, paddy pond mud group.

**Figure 3 ijms-24-14098-f003:**
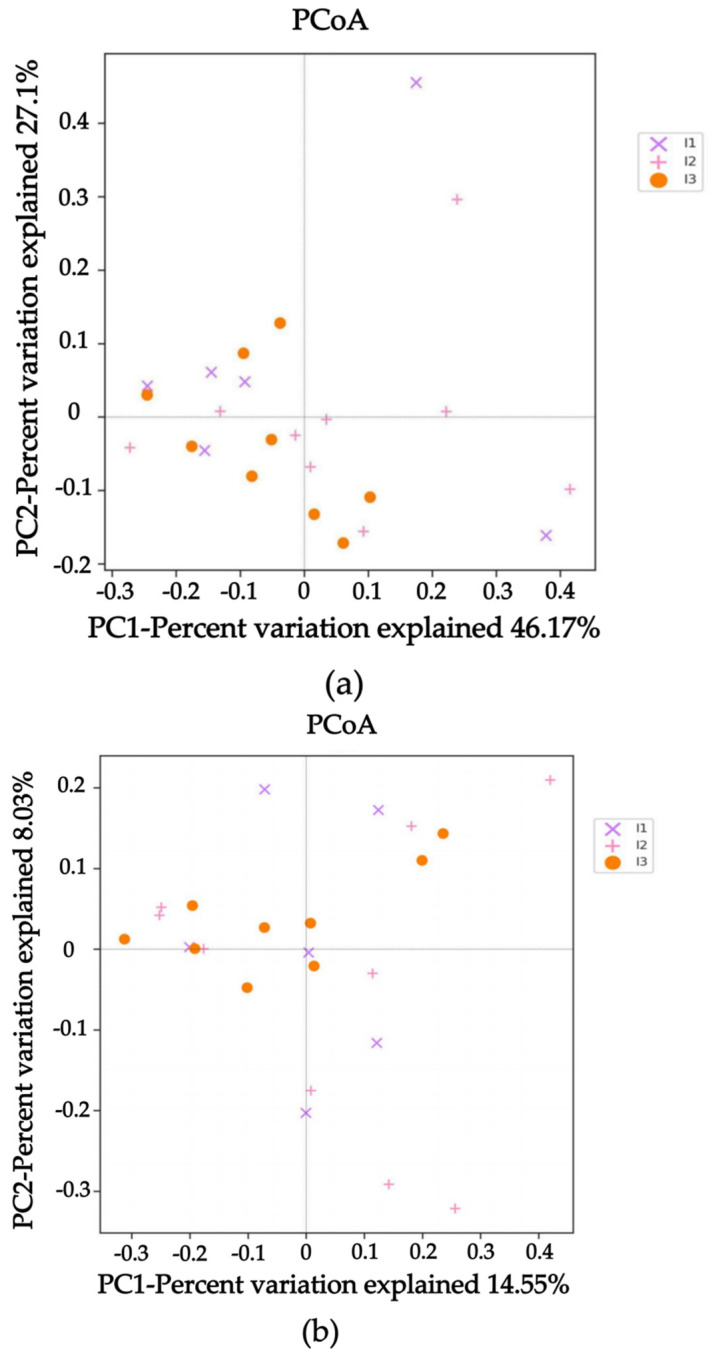
PCoA analysis chart. (**a**) Weighted Unifrac Distance; (**b**) Unweighted Unifrac Distance. I1, crab pond mud group; I2, crayfish pond mud group; I3, paddy pond mud group.

**Figure 4 ijms-24-14098-f004:**
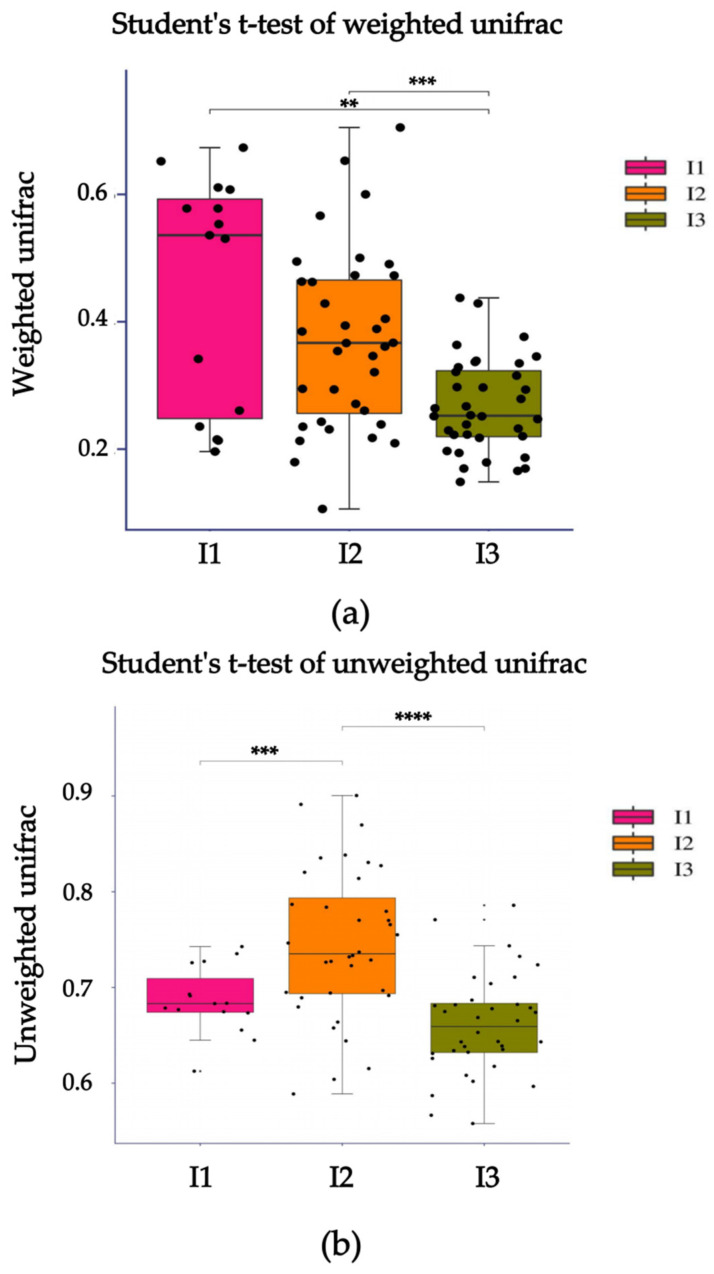
Boxplot of differences in distances between beta diversity groups. (**a**) Weighted Unifrac Distance; (**b**) Unweighted Unifrac Distance. I1, crab pond mud group; I2, crayfish pond mud group; I3, paddy pond mud group. The abscissa is the name of each group, and the ordinate is the value of the corresponding Beta diversity distance index. In the box plot, the meanings of each symbol are as follows: The upper and lower end lines of the bin: upper and lower quartiles (Interquartile range, IQR); Median line: median; Upper and lower edges: maximum and minimum inner circumference values (1.5 times IQR); Outer points on the upper and lower edges: represent outliers. The symbol “**”, “***”, “****” indicate that the difference between the two groups is very significant (*p* < 0.01, *p* < 0.001, *p* < 0.0001).

**Figure 5 ijms-24-14098-f005:**
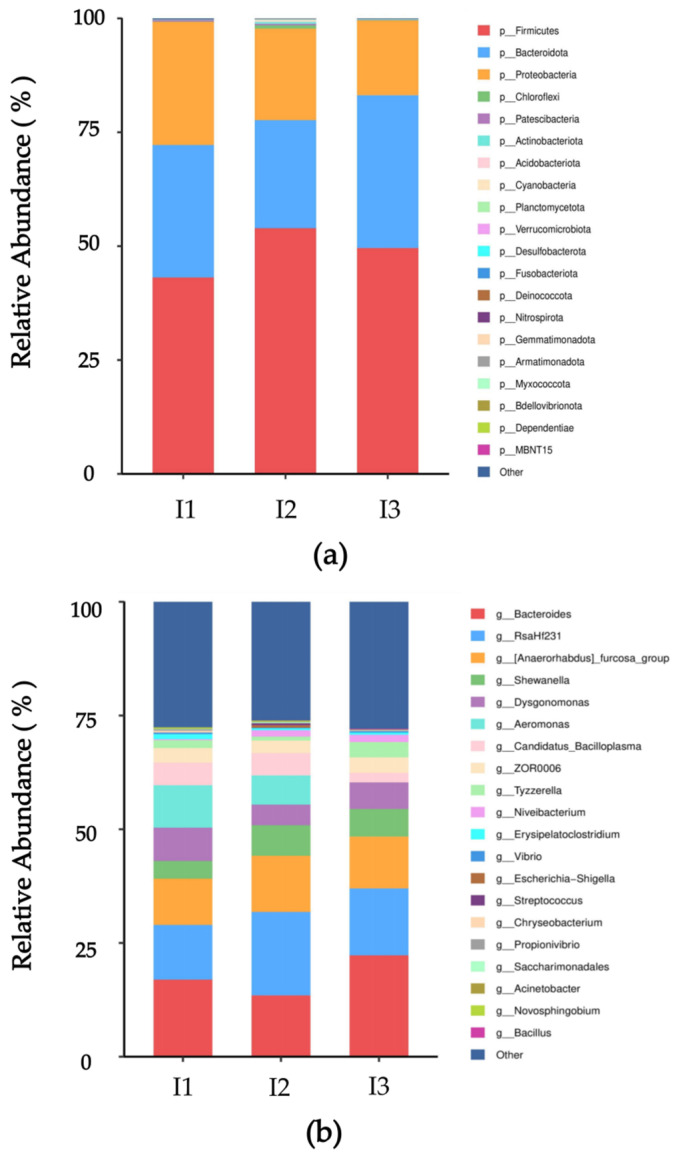
Microflora abundance of intestinal microorganisms in crayfish cultured in substrate sourced from the bottom mud of three groups. (**a**) Phylum level; (**b**) genus level. I1, crab pond mud group; I2, crayfish pond mud group; I3, paddy pond mud group.

**Figure 6 ijms-24-14098-f006:**
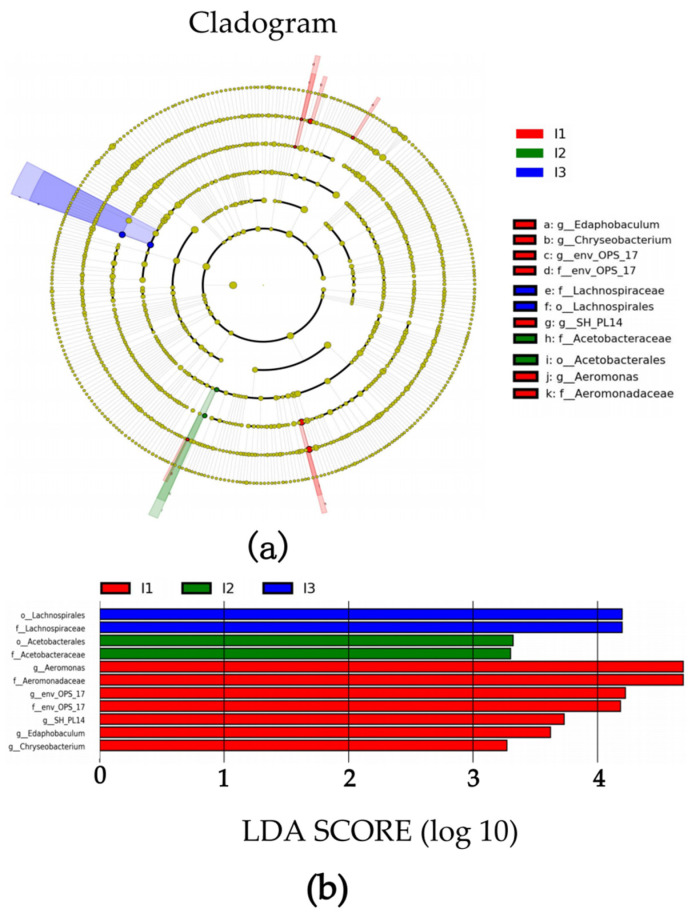
LEfSe analysis chart. (**a**) Clade map of gut microbiota; (**b**) column chart of LDA value. I1, crab pond mud group; I2, crayfish pond mud group; I3, paddy pond mud group.

**Figure 7 ijms-24-14098-f007:**
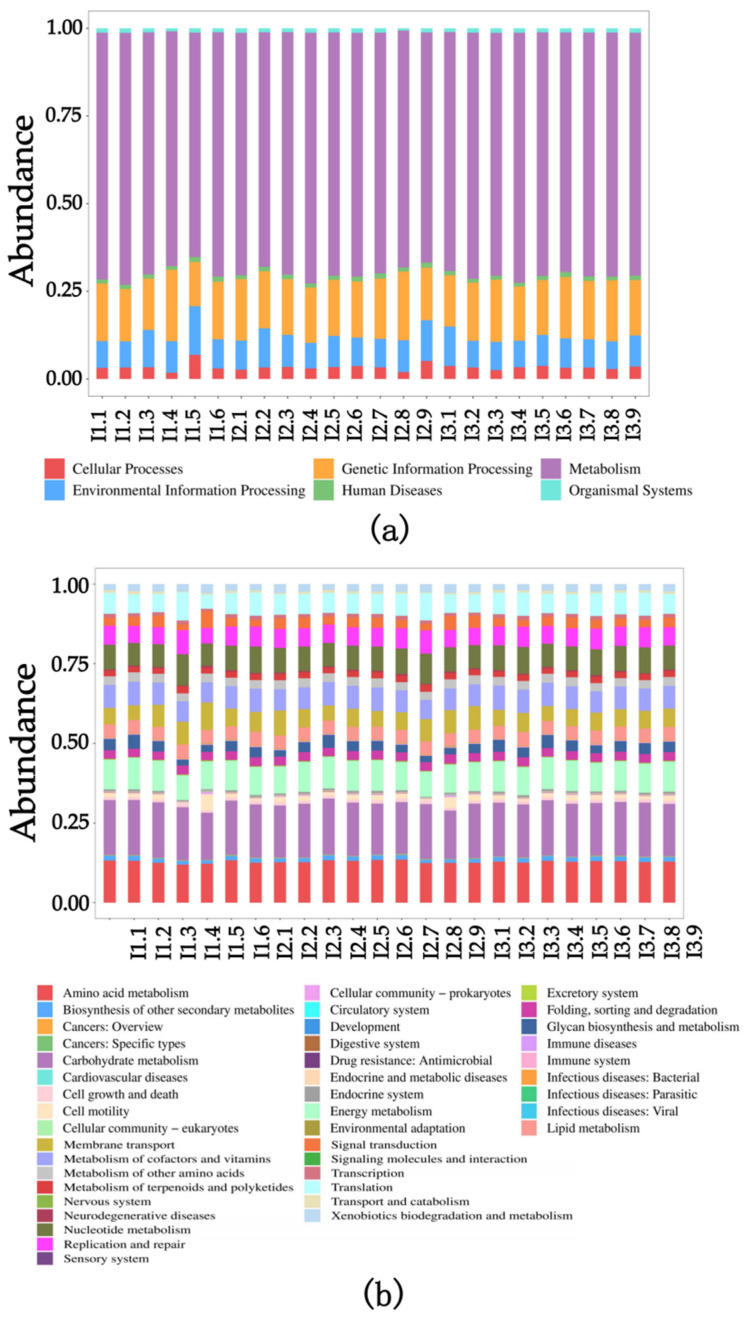
The relative abundance and composition of functional classes predicted by the intestinal flora. (**a**) Primary access; (**b**) secondary pathway. I1, crab pond mud group; I2, crayfish pond mud group; I3, paddy pond mud group.

**Figure 8 ijms-24-14098-f008:**
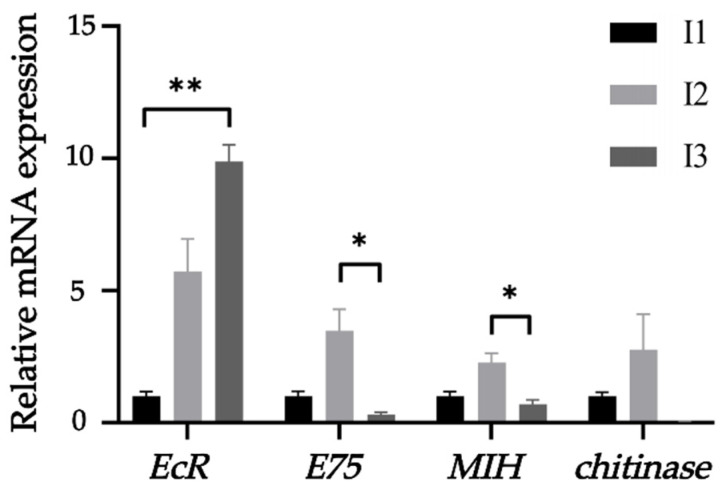
Expression of growth-related genes in hepatopancreatic tissues of crayfish under different substrate conditions. *EcR*, ecdysone receptor; *E75*, nuclear hormone receptor; *MIH*, molt-inhibiting hormone. I1, crab pond mud group; I2, crayfish pond mud group; I3, paddy pond mud group. The symbol “*” on the line between the bars indicates that there is a significant difference between the two groups (*p *< 0.05); “**” indicate that the difference between the two groups is very significant (*p* < 0.01), if *p* > 0.05, it is not displayed by default.

**Table 1 ijms-24-14098-t001:** Effects of different culture substrates on the growth performance of crayfish.

Group	I1	I2	I3
Initial weight/g	8.81 (8.13, 11.39)	9.45 (8.73, 9.94)	10.15 (9.58, 10.67)
Final weight/g	14.27 (14.23, 14.33)	15.71(14.53, 15.53)	13.17 (12.13, 13.62)
WGR (%)	63.03 ^a^ (51.00, 75.03)	66.09 ^a^ (46.18, 77.89)	29.14 ^b^ (18.92, 42.17)
Initial length/cm	7.24 (7.06, 7.41)	7.39 (7.12, 7.62)	7.51 (7.36, 7.75)
Final length/cm	8.37 (8.31, 8.41)	8.59 (8.40, 8.89)	8.26 (8.07, 8.41)
BLGR (%)	15.68 ^a^ (12.15, 19.12)	16.08 ^a^ (10.76, 19.65)	9.86 ^b^ (6.71, 14.27)
SGR (%)	0.97 ^a^ (0.82, 1.12)	1.02 ^a^ (0.76, 1.15)	0.51 ^b^ (0.35, 0.70)
SR (%)	65.00 ^a^ (60.00, 80.00)	73.33 ^a^ (60.00, 80.00)	80.00 ^a^ (70.00, 90.00)

WGR, weight gain rate; BLGR, body length growth rate; SGR, specific growth rate; SR, survival rate; I1, crab pond mud group; I2, pond mud group; I3, paddy pond mud group. The minimum and the maximum values for each parameter are provided in the brackets. Values in the same row with different letter superscripts differ significantly (*p* < 0.05). Values in the same row with the same letter or no letter superscripts same significantly (*p* > 0.05). The same as below.

**Table 2 ijms-24-14098-t002:** Intestinal flora richness and alpha diversity index of crayfish.

Sample	Chao1	Shannon	Simpson
I1	133.67 ± 23.48	5.08 ± 0.57	0.93 ± 0.03
I2	134.88 ± 24.73	5.16 ± 0.28	0.95 ± 0.01
I3	112.13 ± 17.16	5.13 ± 0.22	0.95 ± 0.01

I1, crab pond mud group; I2, crayfish pond mud group; I3, paddy pond mud group.

**Table 3 ijms-24-14098-t003:** Correlation analysis of growth indexes and immune indexes of crayfish in each group.

Index	WGR	BLGR	SGR	SR	ACP	AKP	CAT	SOD
WGR	1							
BLGR	1.00 *	1						
SGR	1.00 **	1.00 *	1					
SR	−0.99	−0.98	−0.99	1				
ACP	0.50	0.48	0.51	−0.64	1			
AKP	0.27	0.25	0.28	−0.42	0.97	1		
CAT	−0.33	−0.31	−0.34	0.48	−0.98	−1.00 *	1	
SOD	−0.23	−0.24	−0.21	0.06	0.73	0.88	−0.85	1

WGR, weight gain rate; BLGR, body length growth rate; SGR, specific growth rate; SR, survival rate; ACP, acid phosphatase; AKP, alkaline phosphatase; CAT, catalase; SOD, superoxide dismutase. Superscript “*” indicates significant correlation (*p* < 0.05) and “**” indicates significant correlation (*p* < 0.01).

**Table 4 ijms-24-14098-t004:** Correlation analysis of growth indexes and growth gene expression of crayfish in each group.

Index	WGR	BLGR	SGR	SR	*EcR*	*E75*	*MIH*	*Chitinase*
WGR	1							
BLGR	1.00 *	1						
SGR	1.00 **	1.00 *	1					
SR	−0.99	−0.98	−0.99	1				
*EcR*	−0.81	−0.82	−0.80	0.69	1			
*E75*	0.72	0.71	0.73	−0.83	−0.17	1		
*MIH*	0.70	0.69	0.71	−0.81	−0.14	1.00 *	1	
*Chitinase*	0.82	0.81	0.83	−0.90	−0.32	0.99	0.98	1

WGR, weight gain rate; BLGR, body length growth rate; SGR, specific growth rate; SR, survival rate; EcR, ecdysone receptor; E75, nuclear hormone receptor; MIH, molt-inhibiting hormone. Superscript “*” indicates significant correlation (*p* < 0.05) and “**” indicates significant correlation (*p* < 0.01).

**Table 5 ijms-24-14098-t005:** Primer sequence of the growth-related genes in crayfish used in the study.

Primer	Sequence (5′-3′)	Usage
*AK*	F: TCCTCGACGTAATCCAGTCC	expression of *AK*
	R: CGAAGTCCTTGTTGGGATGT	
*EcR*	F: GCTCGGACGCAGAGATTCAA	expression of *EcR*
	R: GAAAGTTTTCGCCGCCGATG	
*E75*	F: TGTCTACGACGCCATTAGGC	expression of *E75*
	R: CGAATCTGCGATGTCCACCT	
*MIH*	F: CTCCCAAGATCACAGCGTCA	expression of *MIH*
	R: CAGTTCAAGGTCGAGTCCCA	
*Chitinase*	F: GTACGATCTGCGAGGCAACT	expression of *chitinase*
	R: CAACACCAGTTTGTCAGCGG	

*AK*, aspartate kinase; *EcR*, ecdysone receptor; *E75*, nuclear hormone receptor; *MIH*, molt-inhibiting hormone.

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
