# Peer review of "Effects of Different Sources of Culture Substrate on the Growth and Immune Performance of the Red Swamp Crayfish (Procambarus clarkii)"

_ijms, 2023, doi:10.3390/ijms241814098_

Round 1

Reviewer 1 Report

The manuscript titled “Effects of different sources of culture soils on growth and immune performance of the Red Crayfish (Procambarus Clarkii)” examines the effect of three culture soils on various performance traits of a crayfish species that is quite important for freshwater aquaculture in China. The manuscript apart from examining various growth indexes, searches for differences in the immune systems, through antioxidant indexes, intestinal flora and gene differences that are associated with growth. The main purpose behind the experiments is very interesting due to the fact that nowadays the need to improve the rearing of various aquaculture species is more demanding than ever. The experimental design is solid, with the interdisciplinarity of the research to be very high. However, there were some issues regarding the manuscript that didn’t make it easy to read. Therefore, this manuscript can be accepted for publication, after revisions that are described below and in the attached manuscript that was provided. Following are major and minor comments:

Major

·       Throughout the manuscript all the sections (introduction, material & methods, etc etc.) were presented in double text, thus increasing significantly its size (the yellow labelled texts are the duplications of the previous paragraphs). This made the review of the manuscript quite long. I would suggest revising the whole text.

·       Although the manuscript examines the different substrates for better growth of the red swamp crayfish, there is no mention, of why the specific substrates are selected. It would be good to include an explanation of why the authors selected these substrates and (maybe) the differences between them.

·       All figures have significant results and information, however, both the legends and the axis labels have small font sizes. For the best experience for the readers, it would be good to increase all the font sizes, whereas the legends should be larger so that the readers could easily keep reading the text, without needing to increase zoom in the manuscript (please examine all the figures in the attached revised manuscript to check for specific comments).

·       In Fig. 3 the manuscript uses principal component analysis for the discrimination of the different groups. However, no information on how that method was applied. It would be good to include in the material and methods section the methodology that was followed for anyone who wants to replicate the methodology. Also, the difference between the Weighted and unweighted Unifrac Distances.

·       In the discussion, although in the manuscript there is a clear comparison of the results with other experiments, at the end of this section there is no clear message for its importance on the aquaculture of this species. I would suggest including a paragraph, in which the results of the present manuscript contribute to the rearing of the red swamp crayfish based on the results that were described, thus promoting the advancement of its rearing. Also, if possible, it would be good to include a few words on what will be the next step of this research to furtherly advance Chinese aquaculture.

Minor

·       In the introduction section, the manuscript names four enzymes that were selected to evaluate the antioxidant capacity. However, no information on the selection is mentioned in-text. I would suggest writing a few things for each enzyme and why these four were selected specifically for the selected experiment.

·       In Table 2, the different groups are presented with their coding (i.e., I1-I3). However, in the text (§3.1) they mention the three substrates without the coding, making it difficult to follow up. I would suggest including in parenthesis the codes of each substrate.

·       In the results section, text that describes the results of Fig. 5 is a bit confusing. I would suggest following the sub-section categorization based on the material and methods categorization.

·       In Fig. 8 the legend has only Chinese characters. They must be translated into English.

Author Response

Response to editor,

We express our sincere gratitude for the valuable feedback provided by the editors and reviewers, which we have duly incorporated into the revised version of our submission. Furthermore, we have collectively decided to designate Dr. Wenjing Li as a co-corresponding author, as originally indicated in the initial submission. This decision is based on Dr. Wenjing Li's significant contributions to this study, encompassing writing, reviewing, editing, data curation, and supervision.

Response to reviewer#1

Major comments to address:

The manuscript titled “Effects of different sources of culture soils on growth and immune performance of the Red Crayfish (Procambarus Clarkii)” examines the effect of three culture soils on various performance traits of a crayfish species that is quite important for freshwater aquaculture in China. The manuscript apart from examining various growth indexes, searches for differences in the immune systems, through antioxidant indexes, intestinal flora and gene differences that are associated with growth. The main purpose behind the experiments is very interesting due to the fact that nowadays the need to improve the rearing of various aquaculture species is more demanding than ever. The experimental design is solid, with the interdisciplinarity of the research to be very high. However, there were some issues regarding the manuscript that didn’t make it easy to read. Therefore, this manuscript can be accepted for publication, after revisions that are described below and in the attached manuscript that was provided. Following are major and minor comments:

Reply: We thank the reviewer for this observation and apologize for the confusion. And the manuscript has been corrected accordingly.

Major

Throughout the manuscript all the sections (introduction, material & methods, etc etc.) were presented in double text, thus increasing significantly its size (the yellow labelled texts are the duplications of the previous paragraphs). This made the review of the manuscript quite long. I would suggest revising the whole text.

Reply: We are sorry for the trouble. And the manuscript has been corrected accordingly.

  • Although the manuscript examines the different substrates for better growth of the red swamp crayfish, there is no mention, of why the specific substrates are selected. It would be good to include an explanation of why the authors selected these substrates and (maybe) the differences between them.

Reply: Thank you for your thoughtful suggestions that you made on our original submission. Refer to the data we provide in the latest submission, shrimp paddy culture and pond culture (including shrimp and crab mixed culture) are the most mainstream crayfish farming methods at present. According to the statistical data in 2022, the proportion of the two is 75.06% and 17.86% respectively. What’s more, Aquaculture pond sediments hold significant ecological importance as habitats that are often subject to disturbance. Therefore, we chose the sediment of these mainstream farming methods as the research object. In addition, Zhang et al.(Zhang, Deng, Cao, et al., 2021) found that crab ponds exhibited notably higher alpha diversity in sediment compared to fish and crayfish ponds. Moreover, the diversity and structure of the microbial community were found to be closely associated with sediment properties. These aroused our interest in using the bottom mud as the experimental variable.

  • All figures have significant results and information, however, both the legends and the axis labels have small font sizes. For the best experience for the readers, it would be good to increase all the font sizes, whereas the legends should be larger so that the readers could easily keep reading the text, without needing to increase zoom in the manuscript (please examine all the figures in the attached revised manuscript to check for specific comments).

Reply: Thank you for your suggestions.We have made changes to the figures and legends  accordingly.

  • In Fig. 3 the manuscript uses principal component analysis for the discrimination of the different groups. However, no information on how that method was applied. It would be good to include in the material and methods section the methodology that was followed for anyone who wants to replicate the methodology. Also, the difference between the Weighted and unweighted Unifrac Distances.

Reply: Thank you for your suggestions. And we have made corresponding additions in the material method.

  • In the discussion, although in the manuscript there is a clear comparison of the results with other experiments, at the end of this section there is no clear message for its importance on the aquaculture of this species. I would suggest including a paragraph, in which the results of the present manuscript contribute to the rearing of the red swamp crayfish based on the results that were described, thus promoting the advancement of its rearing. Also, if possible, it would be good to include a few words on what will be the next step of this research to furtherly advance Chinese aquaculture.

Reply: Thank you for your suggestions. The discussion has already been updated with that information.

Minor

  • In the introduction section, the manuscript names four enzymes that were selected to evaluate the antioxidant capacity. However, no information on the selection is mentioned in-text. I would suggest writing a few things for each enzyme and why these four were selected specifically for the selected experiment.

Reply: Thank you for your suggestions. And we have added the basis for selecting these enzymes in the introduction part.

  • In Table 2, the different groups are presented with their coding (i.e., I1-I3). However, in the text (§3.1) they mention the three substrates without the coding, making it difficult to follow up. I would suggest including in parenthesis the codes of each substrate.

Reply: Thank you for your suggestions. And we have made the corresponding additions in the text (§3.1) , and added the footnote.

  • In the results section, text that describes the results of Fig. 5 is a bit confusing. I would suggest following the sub-section categorization based on the material and methods categorization.

Reply: Thank you for your suggestions. And we have rewrote the description of the results of Fig. 5.

  • In Fig. 8 the legend has only Chinese characters. They must be translated into English.

Reply: Sorry to the confusion and we have translated the chinese characters into English in Fig. 8.

Reviewer 2 Report

Review for the paper “Effects of different sources of culture soils on growth and immune performance of the Red Swamp Crayfish (Procambarus Clarkii).”

In this paper, the authors have undertaken a laboratory study with the aim of investigating the impact of diverse substrates on both the growth and the immune performance of juvenile Red Swamp Crayfish. The results revealed that the growth and immune response indices varied according to the type of substrate used. Nevertheless, it was found that the use of pond mud as a growing substrate yielded the most favorable results. Standard procedures were followed while examining the immune response indices and the gene expression. But the report falls short because it omits certain crucial data, particularly regarding the properties of the substrates used in the research. This deficiency makes the study scarcely repeatable. The lack of novelty and a poor discussion only compounds the issue. The presentation too leaves much to be desired. Each paragraph seems to have been duplicated, which not only makes the text challenging to read, but also suggests that the authors did not take the time to review and proofread their work before submitting it.

These mentioned flaws substantially undermine the value of the paper authored by Rong Wan and co-authors and presented to the "International Journal of Molecular Sciences".

Major concerns:

Abstract.

The Abstract needs to be revised. It comes across as an amalgamation of two identical parts. I recommend retaining only the part that commences with "To optimize the culture soil of…” Moreover, all abbreviations utilized in the Abstract ought to be defined.

Introduction.

When it comes to the Introduction, the opening line needs revision. The phrase "The arthropod phylum Crustacean crayfish" strikes me as being odd. The function of the measurement unit "mu’, utilized in crayfish farming production, also needs to be elucidated. Additionally, the first two paragraphs, which are exact copies of each other, needs to be revised to remove redundancy. The text warrants the same treatment for the subsequent paragraphs, which all start with "The substrate…” The authors need to shed more light on what makes their study novel and its relevance to the industry.

Material and Methods.

Regarding the "Methods" section, what do the authors mean when they use the term "uniform physique"? Are they referring to crayfish of identical size? Also, the term "soil" used by the authors seems ill-suited in this discussion, with either "substrate" or "sediment" perhaps better fitting the context. The issue of repeated text in this section mirrors the problem identified in the overall manuscript and needs rectifying.

In addition, crucial information regarding the conditions of rearing during the experiments must be provided. This includes temperature, lighting conditions, oxygen, pH level, types of feed used, feeding frequency, and so forth. More detailed data are required on the substrates used in the study, including grain size, biochemical composition, and other essential parameters.

For the statistical analysis, a parametric approach was adopted by the authors, specifically the one-way ANOVA. However, the prerequisites for employing this method—data possessing normal distribution and homogeneity of variances—were not addressed in the text. Authors are strongly urged to examine their data for normality and heteroscedasticity, apply transformations if necessary; or, if need be, ensue non-parametric methodologies. The methods used for post-hoc testing between different groups needs to be indicated as well.

Lastly, the correlation analysis method used by the authors to establish relationships between growth indices and gene expression should be described in detail given it was not mentioned in the methods section.

Results.

The information contained in Table 2 shows high standard errors in certain instances. The authors are advised to provide the minimum and the maximum values for each parameter. The unexplained abbreviations, specifically I1, I2 and I3, included in the table need to be clarified in an accompanying footnote.

The abbreviations in Figures 1 and 3 require clear elaborations. In the case of Figures 6 and 7, the current quality is quite inferior, necessitating improvement, especially the font size. It has been observed that Figure 8 contains Chinese text. For the benefit of all readers, it would be appreciated if this text were translated into English. The symbols included in the figure also prove challenging to decipher and demand revision.

Tables 4 and 5 are lacking footnotes that explicate all the abbreviations used. It is noticed that the data presented in Tables 4 and 5 closely resemble that of Figure 9, indicating redundancy that should be eliminated.

Discussion.

Upon reviewing the Discussion, it is evident that the text contained in the initial five paragraphs does not introduce any new insights or provide a detailed discourse on the data collected by the authors. Consequently, only the sentence "In this particular investigation, the activities of antioxidant enzymes, namely ACP, AKP, CAT, and SOD, were measured in the serum of Procambarus clarkii grown under three different soil conditions. The results indicated that the immune performance of the serum in the pond mud group outperformed that of the other two experimental groups." should be retained as framing the beginning of the discussion.

The text within the larger paragraph commencing with "Current studies…" confronts a similar issue. Unfortunately, the authors have not provided a discussion on the results regarding the growth performance indices and survival. An equally curt discussion of other results fails to provide new insights into the topic.

The novelty, significance, and applicability of these results within the industry remain vague. Remarkably, the authors have not provided an explanation on how their results can be implemented within the aquaculture of mature animals.

The authors need to remain abreast with recent literature and accordingly update their paper. Currently, it contains a mere 39 sources, of which the necessity to cite the majority is doubtful.

On a concluding note, due to the authors' failure to provide details about the substrate properties used in their study, their work cannot be adequately replicated or applied by industry professionals engaged in aquaculture. The shortcomings indicated above need to be rectified for the research to be considered of substantial value.

Redundant text should be removed.

Author Response

Response to reviewer#2

In this paper, the authors have undertaken a laboratory study with the aim of investigating the impact of diverse substrates on both the growth and the immune performance of juvenile Red Swamp Crayfish. The results revealed that the growth and immune response indices varied according to the type of substrate used. Nevertheless, it was found that the use of pond mud as a growing substrate yielded the most favorable results. Standard procedures were followed while examining the immune response indices and the gene expression. But the report falls short because it omits certain crucial data, particularly regarding the properties of the substrates used in the research. This deficiency makes the study scarcely repeatable. The lack of novelty and a poor discussion only compounds the issue. The presentation too leaves much to be desired. Each paragraph seems to have been duplicated, which not only makes the text challenging to read, but also suggests that the authors did not take the time to review and proofread their work before submitting it.

Reply:Thank you for your constructive comments.And we have modified the corresponding text.

These mentioned flaws substantially undermine the value of the paper authored by Rong Wan and co-authors and presented to the "International Journal of Molecular Sciences".

Major concerns:

Abstract.

The Abstract needs to be revised. It comes across as an amalgamation of two identical parts. I recommend retaining only the part that commences with "To optimize the culture soil of…” Moreover, all abbreviations utilized in the Abstract ought to be defined.

Reply: Thank you for your suggestions. And we have made corresponding additions in the abstract. Moreover, all abbreviations utilized in the abstract has been defined.

Introduction.

When it comes to the Introduction, the opening line needs revision. The phrase "The arthropod phylum Crustacean crayfish" strikes me as being odd. The function of the measurement unit "mu’, utilized in crayfish farming production, also needs to be elucidated. Additionally, the first two paragraphs, which are exact copies of each other, needs to be revised to remove redundancy. The text warrants the same treatment for the subsequent paragraphs, which all start with "The substrate…” The authors need to shed more light on what makes their study novel and its relevance to the industry.

Reply: Thank you for your suggestions. And we have  revised it accordingly and elaborated more on the novelty and its relevance to the industry of the study.

Material and Methods.

Regarding the "Methods" section, what do the authors mean when they use the term "uniform physique"? Are they referring to crayfish of identical size? Also, the term "soil" used by the authors seems ill-suited in this discussion, with either "substrate" or "sediment" perhaps better fitting the context. The issue of repeated text in this section mirrors the problem identified in the overall manuscript and needs rectifying.

Reply: Thank you for your suggestions. And in the newly submitted text, we have unified the use of the word "substrate".

In addition, crucial information regarding the conditions of rearing during the experiments must be provided. This includes temperature, lighting conditions, oxygen, pH level, types of feed used, feeding frequency, and so forth. More detailed data are required on the substrates used in the study, including grain size, biochemical composition, and other essential parameters.

Reply: Thank you for your suggestions. And we have provided the details in the text. (2.1 culture experiment)

For the statistical analysis, a parametric approach was adopted by the authors, specifically the one-way ANOVA. However, the prerequisites for employing this method—data possessing normal distribution and homogeneity of variances—were not addressed in the text. Authors are strongly urged to examine their data for normality and heteroscedasticity, apply transformations if necessary; or, if need be, ensue non-parametric methodologies. The methods used for post-hoc testing between different groups needs to be indicated as well.

Reply: Thank you for your suggestions. And in the newly submitted text, we describe the statistical methods in more detail. (2.8 statistical analysis)

Lastly, the correlation analysis method used by the authors to establish relationships between growth indices and gene expression should be described in detail given it was not mentioned in the methods section.

Reply: Thank you for your suggestions. And in the newly submitted text, we describe the statistical methods in more detail.(2.8 statistical analysis)

Results.

The information contained in Table 2 shows high standard errors in certain instances. The authors are advised to provide the minimum and the maximum values for each parameter. The unexplained abbreviations, specifically I1, I2 and I3, included in the table need to be clarified in an accompanying footnote.

Reply: Thank you for your suggestions. And we have re-verified the statistics and provided the maximum and minimum values for each parameter in Tab.2. In addition, footnotes have been added to explain abbreviations.

The abbreviations in Figures 1 and 3 require clear elaborations. In the case of Figures 6 and 7, the current quality is quite inferior, necessitating improvement, especially the font size. It has been observed that Figure 8 contains Chinese text. For the benefit of all readers, it would be appreciated if this text were translated into English. The symbols included in the figure also prove challenging to decipher and demand revision.

Reply: Thank you for your suggestions.We have made changes to the figures and legends  accordingly.

Tables 4 and 5 are lacking footnotes that explicate all the abbreviations used. It is noticed that the data presented in Tables 4 and 5 closely resemble that of Figure 9, indicating redundancy that should be eliminated.

Reply: Thank you for your suggestions. Footnotes have been added to explain abbreviations. Moreover, we removed Fig.9 to eliminate redundancy.

Discussion.

Upon reviewing the Discussion, it is evident that the text contained in the initial five paragraphs does not introduce any new insights or provide a detailed discourse on the data collected by the authors. Consequently, only the sentence "In this particular investigation, the activities of antioxidant enzymes, namely ACP, AKP, CAT, and SOD, were measured in the serum of Procambarus clarkii grown under three different soil conditions. The results indicated that the immune performance of the serum in the pond mud group outperformed that of the other two experimental groups." should be retained as framing the beginning of the discussion.

Reply: We are sorry for the trouble. And in the revised text, we have made corresponding changes.

The text within the larger paragraph commencing with "Current studies…" confronts a similar issue. Unfortunately, the authors have not provided a discussion on the results regarding the growth performance indices and survival. An equally curt discussion of other results fails to provide new insights into the topic.

Reply: Thank you for your suggestions. In the revised text, we had a deeper discussion so that we could incorporate your new insights.

The novelty, significance, and applicability of these results within the industry remain vague. Remarkably, the authors have not provided an explanation on how their results can be implemented within the aquaculture of mature animals.

Reply: Thank you for your suggestions. And we have made corresponding changes in the discussion process.

The authors need to remain abreast with recent literature and accordingly update their paper. Currently, it contains a mere 39 sources, of which the necessity to cite the majority is doubtful.

Reply: Thank you for your suggestions. And we have corrected and supplemented the references.

On a concluding note, due to the authors' failure to provide details about the substrate properties used in their study, their work cannot be adequately replicated or applied by industry professionals engaged in aquaculture. The shortcomings indicated above need to be rectified for the research to be considered of substantial value.

Reply: We express our gratitude for the valuable suggestions provided, and we have duly incorporated comprehensive information regarding the characteristics of the substrates employed in the investigation. Nevertheless, we respectfully hold a divergent viewpoint from the reviewer's assertion that "the work lacks sufficient reproducibility or applicability for aquaculture practitioners in the industry." In response, we have included a corresponding discourse within the revised manuscript.

Round 2

Reviewer 2 Report

Second review for the paper “Effects of different sources of culture soils on growth and immune performance of the Red Swamp Crayfish (Procambarus Clarkii).”

The authors have meticulously executed alterations in the manuscript, notably refining the textual content and enhancing the visual representations of the data.

Nonetheless, a pivotal issue related to their experimental technique continues to persist, unresolved.

1) The current abstract is excessively lengthy and should be revised and shortened.

2) Addressing the specifics, Figures 5 through 7 exhibit deficient resolution, limiting the clarity of the data they aim to portray.

3) A critical lapse in information provision by the authors is evident. They have failed to disclose the chemical composition and microbiota of the three different substrate types utilized during the experiment. This omission necessitates the reiteration of my previous conclusion: the significance of these results and their potential applicability within the industry retain a certain level of ambiguity. I pose the question to the authors - can they provide assurance that the sediments employed in their experiments possess similar qualities as those found in other ponds/regions? In the absence of any supporting data for this assumption, other aquaculture practitioners find themselves unable to enact these findings practically. 

Minor

Author Response

Response to reviewer#3

1) The current abstract is excessively lengthy and should be revised and shortened.

Reply: We thank the reviewer for this observation and apologize for the confusion. And the abstract has been revised again and condensed.

2) Addressing the specifics, Figures 5 through 7 exhibit deficient resolution, limiting the clarity of the data they aim to portray

Reply: We thank the reviewer for this observation and apologize for the confusion. We have resubmitted a clearer picture.

3) A critical lapse in information provision by the authors is evident. They have failed to disclose the chemical composition and microbiota of the three different substrate types utilized during the experiment. This omission necessitates the reiteration of my previous conclusion: the significance of these results and their potential applicability within the industry retain a certain level of ambiguity. I pose the question to the authors - can they provide assurance that the sediments employed in their experiments possess similar qualities as those found in other ponds/regions? In the absence of any supporting data for this assumption, other aquaculture practitioners find themselves unable to enact these findings practically.

Reply: We thank the reviewer for this observation and apologize for the confusion. In this experiment, the water quality index and a series of chemical components are in the normal range. The effects of Procambarus clarkii on organic matter degradation and bacterial community in surface sediments can be referred to the published literatures (Hou Y, Jia R, Ji P, Li B, Zhu J. Organic matter degradation and bacterial communities in surface sediment influenced by Procambarus clarkia. Front  Microbiol. 2022 Oct 21; If 85555. Doi: 10.3389 / fmicb. 2022.985555. PMID: 36338081; PMCID: PMC9634481.); The effects of culture on the dynamics and aggregation of benthic bacterial communities in paddy fields were currently based on the reference journal environments. Authors Yiran Hou, Rui Jia, Wei Sun, Houmeng Ding, Bing Li *, and Jian Zhu (in the throw); For the effect of shrimp and crab mixed soil flora, we refer to the article《Is rice-crayfish co-culture a better aquaculture model》.

Round 3

Reviewer 2 Report

Third review for the paper “Effects of different sources of culture soils on growth and immune performance of the Red Swamp Crayfish (Procambarus Clarkii).”

The authors have revised the paper, but further revisions are needed.

1) The authors mention "Xitangni group, crayfish Tangni group and Shuitangni group" in the abstract, but no description is given for them. This information is also missing in the text.

2) The authors provided three references in their response to my concerns, but none of them are visible in the text and reference list. An update is needed to provide information on the microbiota and chemical composition of the substrates used in the experiments and, as the authors state, referenced in these sources.

3) The authors should strictly adhere to the convention of italicizing Latin names in the reference section.

Some revisions are required.

Author Response

Response to reviewer#4

1)  The authors mention "Xitangni group, crayfish Tangni group and Shuitangni group" in the abstract, but no description is given for them. This information is also missing in the text.

Reply: We thank the reviewer for this observation and apologize for the confusion. Each breeding pond has three years of breeding experience before, feeding normally, no disease, and water quality testing is often carried out, which meets the standard requirements. I have added in the material.

2) The authors provided three references in their response to my concerns, but none of them are visible in the text and reference list. An update is needed to provide information on the microbiota and chemical composition of the substrates used in the experiments and, as the authors state, referenced in these sources.

Reply: We thank the reviewer for this observation and apologize for the confusion. The chemical composition of the three substrates is at the normal level, and the composition of the microbial community is mainly Firmicutes, Proteobacteria, Bacteroidetes, Chloroflexi, Acidobacteria, Actinobacteria, chloroflexi. Patescibacteria, Verrucomicrobia, and BRC1.( Hou Y, Jia R, Ji P, Li B, Zhu J. Organic matter degradation and bacterial communities in surface sediment influenced by Procambarus clarkia. Front Microbiol. 2022 Oct 21;13:985555. doi: 10.3389/fmicb.2022.985555. PMID: 36338081; PMCID: PMC9634481.)

3) The authors should strictly adhere to the convention of italicizing Latin names in the reference section.

Reply: We thank the reviewer for this observation and apologize for the confusion. Changes have been made.